# Identification and Mitigation of Subsidence in Karstic Areas with Sustainable Geotechnical Structures: A Case Study in Gallur (Spain)

Alberto Gracia [1,*], Francisco Javier Torrijo [2], Julio Garzón-Roca [3] and Miguel Pérez-Picallo [1]

[1] C.T.A. Associated Technical Consultants, S.A.P., 50006 Zaragoza, Spain; mperez@cta-consultores.com
[2] Research Centre for Architecture, Heritage and Management for Sustainable Development (PEGASO), Department of Geotechnical Engineering, Universitat Politècnica de València, Camino de Vera s/n, 46022 Valencia, Spain; fratorec@trr.upv.es
[3] Department of Geodynamics, Stratigraphy and Paleontology, Faculty of Geology, Complutense University of Madrid, 28040 Madrid, Spain; julgarzo@ucm.es
* Correspondence: agracia@cta-consultores.com; Tel.: +34-677480066

**Abstract:** In various areas of the Ebro valley in Spain, including the region discussed here, the risk of sinkholes is becoming particularly severe, particularly impacting urban areas and roadways where land subsidence from karstic processes is common. However, knowledge of the area, its geological–geotechnical configuration, and the carrying out of specific research studies are allowing solutions to be tested in an attempt to resolve these situations. A case in point is the examination of settlement issues along a stretch of the access road leading to the city of Gallur from the east (known as Camino Real) in the Zaragoza province, Spain. Numerous surface manifestations of recent subsidence and/or collapse activities have been observed, manifesting as craters and ground undercuts, some several meters in diameter. The prevalence of highly karstifiable materials in the area, evident from the existence of subsidence pockets and collapse dolines, poses significant safety concerns, particularly for traffic and town access, prompting the closure of Camino Real for several years. Local and provincial authorities have embarked on studies to try to recognise this type of situation. Reports aimed at defining karstification processes, conducting geomechanical analyses of subsidence and cavity collapses, and proposing technical measures to mitigate risks have been prepared. Finally, a consolidation solution was proposed based on injections at column-depth of mortar with special characteristics, combined with the replacement and reinforcement of the most superficial soil by means of high-tensile-strength geotextile meshes.

**Keywords:** hazard; sinkhole mitigation; gypsum karst; Gallur; injections; geogrid; evaporitic rocks





## 1. Introduction

A sinkhole is a geological hazard characterized by a depression or cavity in the ground. The risk of sinkholes is escalating, particularly in urban areas lacking meticulous planning, where karst depressions are frequently filled and developed. Effectively addressing these risks requires the identification, investigation, prediction, and mitigation of sinkholes [1]. Corrective measures can be implemented to mitigate subsidence processes.

As said, subsidence and sinkholes pose a particularly significant challenge in urban areas situated within karstic regions [2,3]. Such regions are identified by the prevalence of soluble rocks, either carbonate rocks or evaporitic ones. These areas encompass approximately 20% of the Earth's ice-free continental surface [4,5]. The dissolution of these soluble strata, forming the rocky substratum, initiates a process of upward subsidence towards the surface, which, in certain instances, results in episodes of collapse [1,6–9]. Sinkholes are more likely to occur and exhibit greater genetic diversity in evaporitic rocks compared to carbonate rocks. This is attributed to the higher solubility of the former, such as halite

and gypsum. The progression of sinkhole development occurs more rapidly in evaporitic rocks, with gypsum being approximately 100 times more soluble than carbonate rocks and this ratio being even higher for halite. Subsidence damage resulting from the dissolution of such rocks leads to substantial global losses. Consequently, the presence of evaporitic rocks poses significant challenges in built environments [10].

Sinkholes emerge sporadically due to the gravitational movement of the overlying material situated above the soluble rock stratum [9,11,12]. The overlying material typically consists of residual soils, referring to in situ disturbed substrate. The gradual dissolution of soluble rocks at depth, resulting from the infiltration and flow of water, gives rise to the formation of cavities or domes near the interface between residual soils and the soluble rock [9,11–13].

Sinkhole hazards exert a significant economic impact in Spain, as indicated by recent studies focused on identifying and investigating sinkholes in the Iberian Peninsula [9,14–16]. Specifically, the city of Zaragoza in northeastern Spain and its surrounding areas are identified as the highest sinkhole risk area in Europe [17]. The abundance of gypsum serves as the primary origin of these sinkholes, although the interstratal dissolution of halite and glauberite beds also contributes to their development [9,17,18]. In this context, this paper presents a case study in Gallur, a town situated in a high-risk area in Spain (Figure 1). Over the course of several years, the access road to Gallur from the east (Camino Real) experienced disruptions due to the emergence of a sinkhole, impeding access to the town. A preliminary examination of the area uncovered a sinking sinkhole, with its most depressed sector located immediately to the southwest. In this region, a cluster of discharge edges was identified, undermining the mentioned road surface (Figure 2).

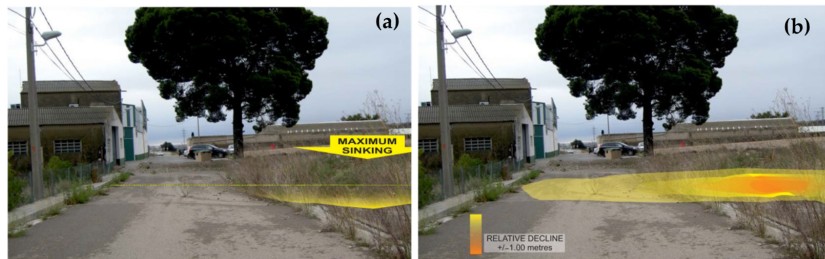

**Figure 1.** Panoramic images of the apparent subsidence of the road surface of the access to Gallur along the Camino Real (2015). This access had remained closed for several years. (**a**) The location of the maximum observed subsidence is shown. (**b**) The most depressed area is highlighted in color.

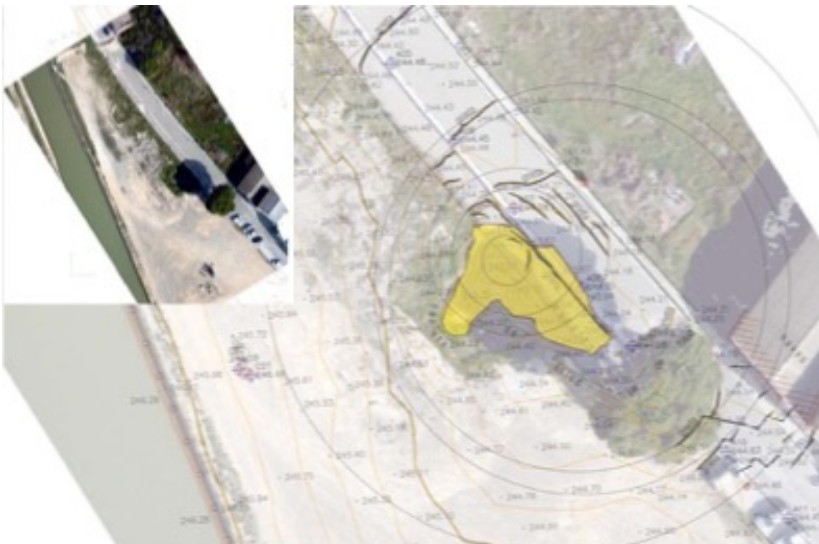

**Figure 2.** Topography via aerial photography (year 2015). The area affected by land subsidence is marked in colour (yellow). The layout of the contour lines shows the subsidence edges and defined

areas affected by structural changes. They can be correlated allowing concentric circles to be formed with the depocenter in a centered position.

## 2. Materials and Methods

### 2.1. Geographical and Geological Framework

Gallur is located in the central western sector of the Ebro Basin, in the province of Zaragoza (Figure 3a,b). The climate is continental, with an average annual temperature between 14 and 15 °C and rainfall of less than 400 mm/year. The temperature fluctuations are strong, with maximum values close to 45 °C and minimum ones below −15 °C.

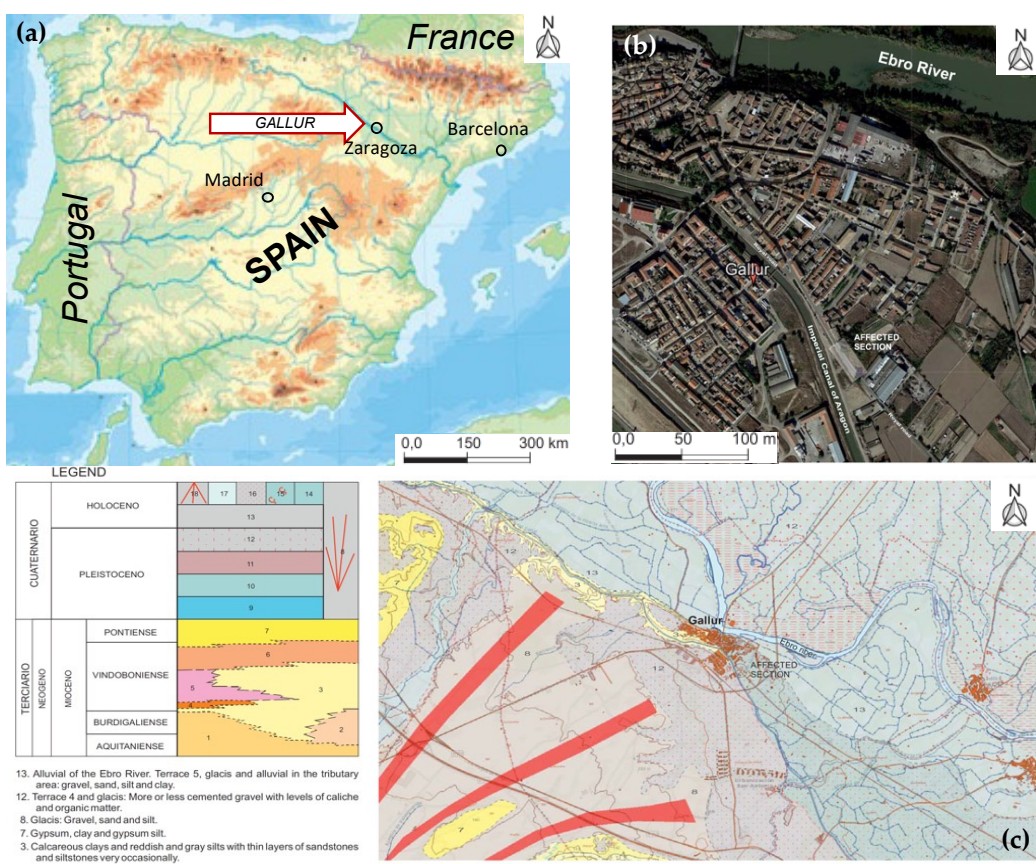

**Figure 3.** Geographical and geological framework: (**a**) Location of Gallur in the Iberian Peninsula; (**b**) Aerial view of the area under study (source: Google Maps); (**c**) Geological map of Alcalá de Ebro and its surroundings (source: [19]).

Geologically, the town is situated on Quaternary materials characterized by variable thickness and compactness, overlying a Tertiary rocky substratum with a loamy gypsiferous composition (Figure 3c). The materials that make up the Tertiary sedimentary column have a rhythmic vertical arrangement. In general, these rhythms show two well-defined members or sections: a basal one of a fundamentally terrigenous character and an upper one of a carbonate and/or evaporitic constitution.

The Quaternary formations overlying the Miocene substratum outcrop extensively throughout the sector. They are arranged in various terrace levels associated with the Ebro, as well as different glacial deposits, colluvia, and valley bottoms. Different types of deformations can be distinguished, which fall into four genetic groups: karstic, diapiric, tectonic, and hydroplastic. Karst deformations are observed as sinform structures, normal faults and flexures, internal unconformities, and so on, and they are the result of the solubilisation of evaporitic materials under Quaternary deposits. In many cases, these morphologies are fossilised by later deposits. At present, their functionality is recognised

by the generation of dolines that affect alluvial terraces in the 30 m and 55 m and Neogene terraces, causing spectacular collapses [7–9,18,20,21].

## 2.2. Geotechnical Investigation and Geotechnical Profile

As a starting point, based on data from previous works [21], boreholes and geophysical research of the existing detailed topography were located (Figure 2). The ground condition research facilitated the creation of a correlation model for the terrain, which was used to set the size and to delineate potential ground consolidation solutions (Figure 4).

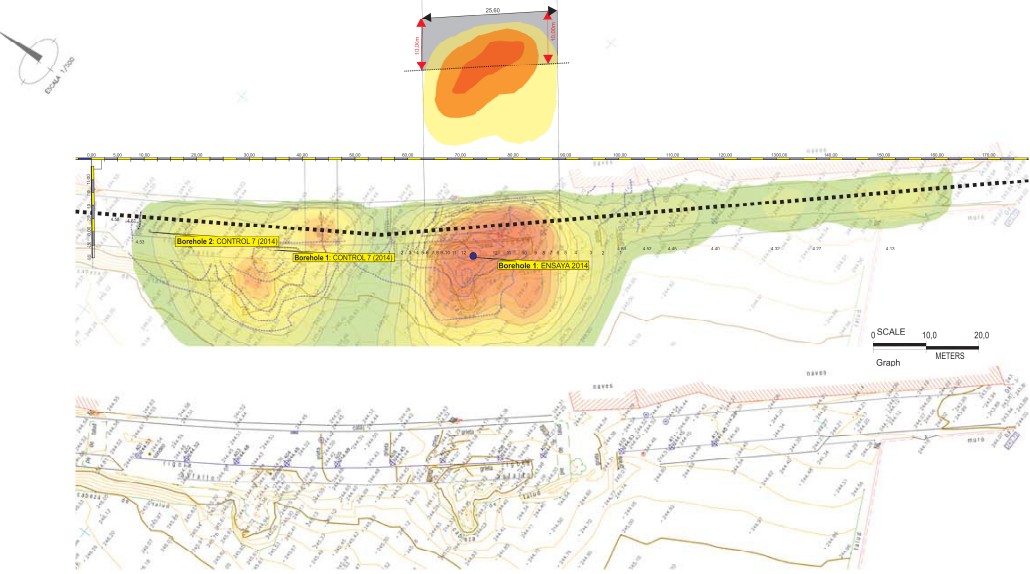

**Figure 4.** Situation plan with positioning of the subsidence model defined, based on the existing data.

These previous works consisted in carrying out two geotechnical boreholes of 16.5 and 18 m, respectively. And, in addition, 54 georadar profiles representing a linear survey distance of 3110.05 m were also carried out.

In the correlation conducted (Figure 5), two reference (guide) levels were obtained, serving to establish the relative decreases recorded in the subsidence zone affected by the sinkholes in this particular case. Level two (2), made up of coarse gravel with sand, showed a relative drop of around 3–4 m in the sounding. Reference level 4 also showed a relative drop of another 4–5 m, so that the accumulated result would be in the order of 7 to 9 m. The thickness of level 3 reached more than 10 m in the area that registers a subsidence process, while in the part where apparently that did not occur, the thickness was between 5 and 6.5 m. One can interpret that the gravel with sand that forms this defined level is very dense and compact ($N_{SPT} > 50$/Rejection) in boreholes 1 and 2 and much looser, not very compact, in the borehole performed in the sinkhole, where the resistance to penetration ($N_{SPT} = 5$–$14$) is significantly lower. The model considered the ground following the geotechnical profile obtained in this work (see Table 1 for the corresponding properties).

**Table 1.** Ground material properties.

| Material | Unit Weight [kN/m³] | Young Modulus [MPa] | Poisson Ratio [-] | Cohesion [kPa] | Friction Angle [°] | Permeability [m/s] |
|---|---|---|---|---|---|---|
| Level 0 (Fills) | 19 | 7 | 0.3 | 10 | 15 | 0.001 |
| Level 1 | 20 | 16 | 0.3 | 20 | 28 | 0.01 |
| Level 2 | 20 | 25 | 0.3 | 30 | 30 | 0.01 |
| Level 3 | 21 | 45 | 0.3 | 40 | 36 | 0.01 |
| Level 4 (Altered substrate) | 21 | 36 | 0.3 | 100 | 29 | $10^{-7}$ |
| Level 5 (Marl–gypsum substrate) | 22 | 80 | 0.3 | 400 | 33 | $10^{-7}$ |

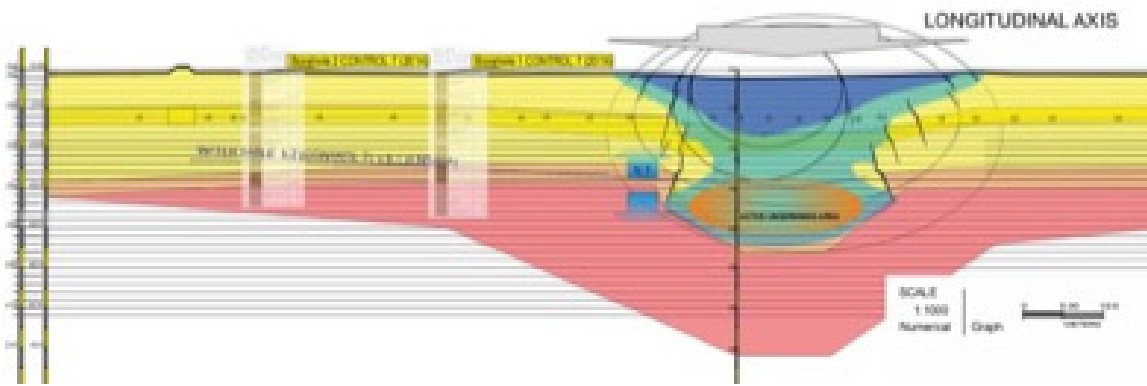

**Figure 5.** Longitudinal geological–geotechnical profile established based on the existing data.

## 3. Results and Discussion

*3.1. Analysis and Interpretation of the Dynamics Involved in the Formation and Evolution of Dolines*

In the investigation located in the center of the area with the highest magnitude of surface settlement, no hollows were found. Beyond the level of brown clays (level 4), there were clays and claystones that constitute an apparently altered, but reasonably continuous and fairly firm, substrate (level 5).

Thus, the materials that form the rocky substrate (levels 4 and 5) on which the alluvial gravels that form the current terrace of the river Ebro (levels 1, 2 and 3) were deposited are particularly susceptible to erosion. Their clayey–loamy, loamy–clayey nature and the frequent occurrence of gypsum content (e.g., nodules, veins, and strata) make them easily undermined in conditions in which favourable factors combine. The possibility of fractures in the rocky substratum and its relative slope towards the river facilitates, in certain areas, the generation of concentrated flows (within the water table) where, taking advantage of this greater permeability, strong internal scour processes are generated, which evolve into hollows or cavities, which finally come to the surface in the form of chasms or sinkholes.

Based on previous studies [1,6–9,22,23], the following scheme can be applied (Figure 6):

1. In the area of the oscillation of the water table and in the contact between the deposit of granular material (gravel and sand) and the clay–marl substratum (with gypsum), there is an anomalous differential concentration of a sub-valve flow in the direction of the river Ebro.
2. A higher velocity flow is generated in this area and, consequently, a greater dragging capacity (scour) of both the fines in the granular deposit and those displaced due to the alteration of the substratum.
3. For this reason, the gravel of level 3, in the zone of contact with the clayey substrate, has a low fines content, which results in a higher relative permeability and abnormally low strength.
4. Hence, the process can be likened to a large hourglass with lateral expansion, favoring the preferential flow line toward the base level of the entire aquifer system, situated in the direction of the Ebro river.

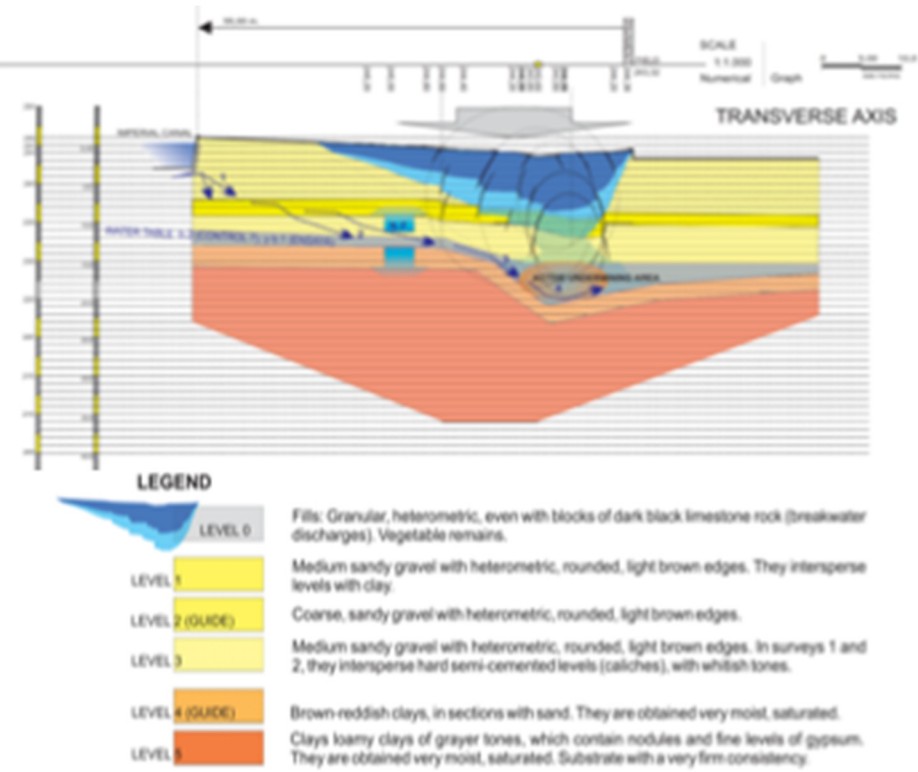

**Figure 6.** Geological–geotechnical cross-sectional profile established based on previous existing data (source: [21]).

### 3.2. Proposed Solution

3.2.1. Approach to the Solution

From a perspective taken from a transversal axis of the road (Figure 6) and based on the surface morphology of the terrain, an interpretation of the situation can be made that leads to a model in which the following aspects can be assessed:

- There is no nearby catchment basin that could justify a process of this magnitude. The unusual influx of water into the system might be attributed to the nearby Canal Imperial de Aragón. In recent history, it is widely known that repairs were necessary on a section located immediately to the west, merely 30–40 m away. The same issue that prompted the aforementioned repair could have influenced the emergence of the new subsidence observed in this access area to Gallur.
- These inputs have been infiltrating for a long time through the gravel deposit that makes up the soil in the area. The infiltrated flow was incorporated into a system of high relative porosity (levels 1 and 2), taking advantage of levels where the proportion of fines is lower (level 3), incorporating itself into the phreatic. The natural, preferential direction of this flow is in the direction of the river Ebro, towards the west.
- Taking into account the correlation among the boreholes surveyed, it is conceivable that there was a reduction in the relief corresponding to the Tertiary substratum. In this relative difference in elevation, there is a shift in the speed of the phreatic flow, which also integrates infiltrated contributions from the nearby channel, likely accumulating over the years. This contact gives rise to a flow with higher velocity and, consequently, an enhanced capacity for dragging (scour) of both fines from the granular deposit itself and those displaced due to the alteration of the substratum.
- Even though the issue of seepages from the canal was addressed at the time, the process retains a certain inertia that promotes the emergence of sinkholes long after the required repairs to the canal were completed. It is crucial to ensure that the canal does not persist in leaking within this environment.

Based on all the aspects analyzed and on the basis of the interpreted geological and geotechnical model, the most suitable solution was assessed. The following arguments were considered:

1.  The process of subsidence in this particular area is immediately related to the problems of seepages that occurred from the Canal Imperial de Aragón from the section that was replaced by an artificial canal–aqueduct that apparently ceased to provide these seepages, which, nevertheless, could have been acting for a long time.
2.  The model, developed using data from the geophysical campaign and borehole surveys, suggests that the ongoing scour process, which led to the land sinking, primarily took place at the interface between the gravelly sands of the alluvial deposit and the loamy clay with gypsum in the underlying substratum.
3.  Although there is a possibility that the process of scour and creep may affect levels or strata located within the rocky substratum, at greater depths, in order to be able to resolve the problem, once the origin of the problem was apparently addressed, acting on the soil of the first levels (on the gravel with sand of the alluvial terrace deposit) is considered appropriate. This would involve filling the voids in the area of the street layout and compacting the soil, to recover as much bearing capacity as possible.

Finally, the solution adopted consisted of consolidating the ground in the section where the settlements were located by injecting low-mobility mortar in the center of the sunken area (Figure 7) and by the construction of an embankment reinforced with high-tensile-strength geogrids along the entire section of the affected street.

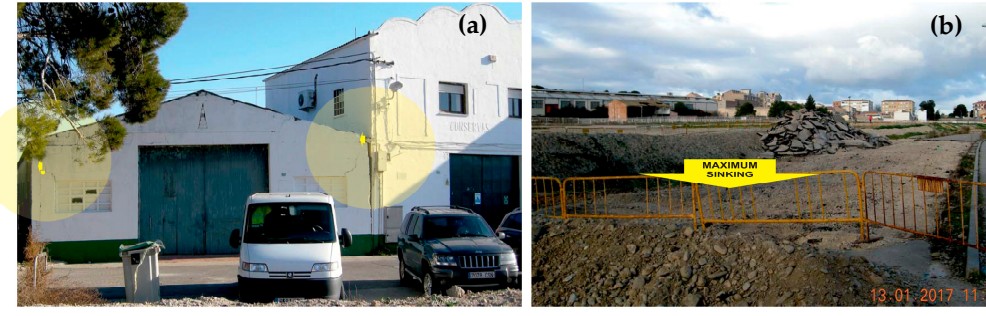

**Figure 7.** (**a**) Damage to an adjacent building is evident, highlighting points where deformations (cracks) are most pronounced, along with their indicated direction. (**b**) View of the street affected by the subsidence where the maximum subsidence is marked.

In the choice of alternative solutions, the possible environmental impacts on the environment were taken into account for the injection procedure. Thus, the procedure is based on the injection of special mortars that are defined as 'low mobility', which means that they remain at the injection point itself without being affected by sub-valve flows that transport them. They are therefore "high viscosity dry mortars" which, by definition, do not mix with the ground, do not impregnate it to reduce its permeability (for example), and do not tend to fracture it. Therefore, they form localised masses that, by virtue of their injection procedure at very high pressures of up to 60 bar, fill possible voids and compact and consolidate the soil where it is looser. For all of the above reasons, it is a material that interacts minimally with its surroundings, and once it has set, it remains inert in its position. In this sense, it is clearly considered sustainable.

In comparison to other techniques, such as more liquid injections or jet grouting-type systems, where much more fluid is used that can be incorporated more into possible underground flows or seepage, the mortars are considered to be much more inert.

3.2.2. Adopted Solution

Injection Works

The injection works were carried out according to the following procedure:

1.  Installation on site and restoration of pavements. Trench excavation, up to −3.00 m, of the area (section of street) in which the consolidation injections were subsequently carried out. In point (a) of Figure 8, the specific area is included in the plan view.
2.  Access and staking out of the injection equipment campaign (Figure 9). A test section was carried out to define the spacing of the injections and to evaluate the injection pressures. DPSH tests were carried out to verify the situation.

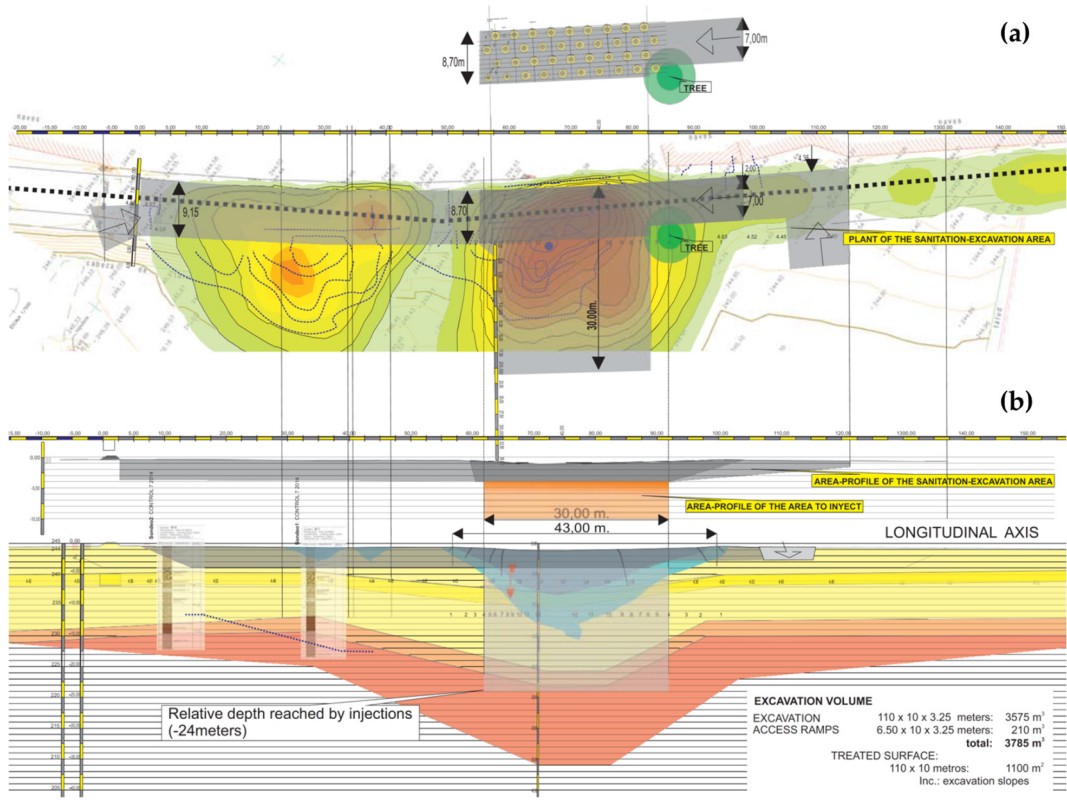

**Figure 8.** Location in plan view and profile of the consolidation works. (**a**) Initial excavation (in plan view) to carry out the consolidation mortar injections. (**b**) Excavation in the second phase (in profile), for the rehabilitation and placement of a reinforced–compacted embankment using high-tensile-strength geogrids. The relative depth reached by the B.M. mortar injections is included in the diagram that serves as a geotechnical model.

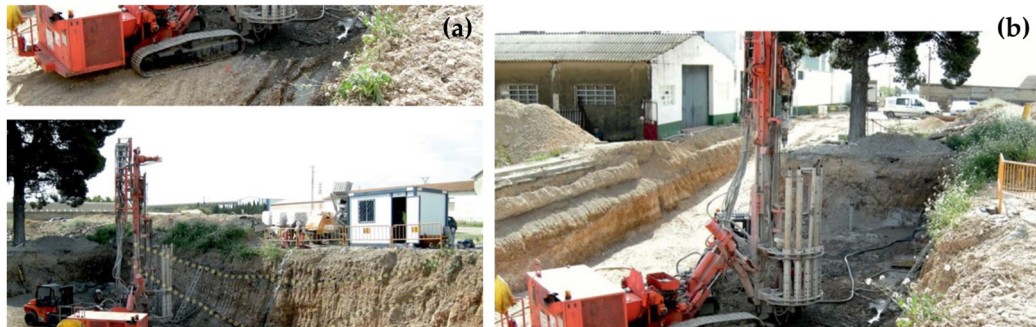

**Figure 9.** (**a**) Drilling equipment located in the excavation (at −3.00 m). (**b**) Excavation slope with the apparent arrangement of the ground levels in the shape of a trough, indicating the point of maximum subsidence.

3.  Execution of low-mobility mortar injections (compaction grouting), in the excavation made (Figure 10), at −3.00 m from the current street level. Maximum planned consumption of 150 L/m, including drilling into the ground. The mortar was injected

at a pressure up to 20 bar and with an Abrams cone value of less than 8 to 12 cm. The pumping speed was limited to less than 60 L/m in order to produce infiltration-displacement of the soil without breaking its structure.

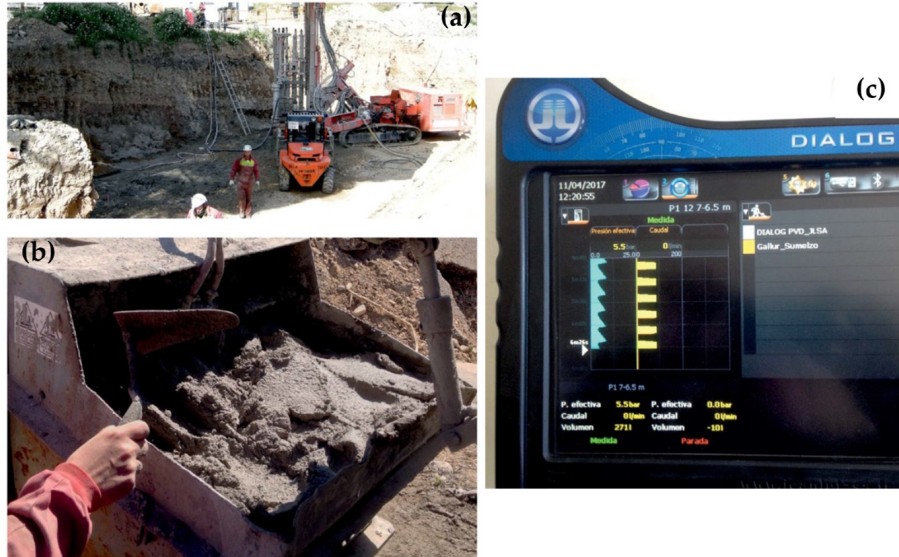

**Figure 10.** (**a**) Drilling equipment located in the excavation made at −3.00 m from the original street surface; (**b**) Appearance (texture and consistency) of the injection mortar in the injection pump hopper; (**c**) Injection control equipment: injection in the interval between −7.00 and −6.50 m; effective pressure at that time 5.5 bar; mortar intake volume, after 6.26 min: 271 l.

The work was carried out in the arrangement shown in Figure 11 and to previously determined depths at each injection point, initially assessed to be −21.00 m deep from the excavation grade, at −3.00 m from the current street.

In order to improve the information on the ground situation at depth, seven of the 38 injections of 29 m were extended to −27 m (Tables 2 and 3).

**Table 2.** General conditions for drilling. Numbers 1 to 38 (4 April 2017 to 13 June 2017).

| | |
|---|---|
| Open circuit pressure | 2 a 5 bars |
| Density mortar | 1.9 kg/L |
| Water content | 18–20% |
| Maximum planned volume (L/mL) | 150 L/mL * |
| Effective pressure (bars) | 10–15 bars ** |

\* If a minimum pressure of 10–15 bars. ** Above pressure of open circuit pressure.

**Table 3.** Drilling data performed.

| | Order | Date | Nº Drilling. | Meters | Litres Registered | Kg Mortar | Litre/mL | Pallets |
|---|---|---|---|---|---|---|---|---|
| **Line 1 (10 uds.)** | 1.00 | 42,829.00 | 3.00 | 21.00 | 2542.12 | 4830.03 | 121.05 | 2.76 |
| | 2.00 | 42,830.00 | 5.00 | 21.00 | 2542.13 | 4830.05 | 121.05 | 2.76 |
| | 3.00 | 42,836.00 | 7.00 | 21.00 | 3094.73 | 5879.99 | 147.37 | 3.36 |
| | 4.00 | 42,837.00 | 9.00 | 21.00 | 2586.22 | 4913.82 | 123.15 | 2.81 |
| | 5.00 | 42,843.00 | 1.00 | 21.00 | 1789.30 | 3399.67 | 85.20 | 1.94 |
| | 6.00 | 42,844.00 | 6.00 | 21.00 | 3912.65 | 7434.04 | 186.32 | 4.25 |
| | 7.00 | 42,845.00 | 10.00 | 21.00 | 2155.29 | 4095.05 | 102.63 | 2.34 |
| | 8.00 | 42,846.00 | 2.00 | 21.00 | 1293.12 | 2456.93 | 61.58 | 1.40 |
| | 9.00 | 42,870.00 | 8.00 | 27.00 | 4156.00 | 7896.40 | 153.93 | 4.51 |
| | 10.00 | 42,879.00 | 4.00 | 27.00 | 3920.54 | 7449.03 | 145.21 | 4.26 |

**Table 3.** *Cont.*

| | Order | Date | Nº Drilling. | Meters | Litres Registered | Kg Mortar | Litre/mL | Pallets |
|---|---|---|---|---|---|---|---|---|
| **Line 2 (9 uds.)** | 11.00 | 42,851.00 | 11.00 | 21.00 | 1724.11 | 3275.81 | 82.10 | 1.87 |
| | 12.00 | 42,859.00 | 19.00 | 21.00 | 2155.21 | 4094.90 | 102.63 | 2.34 |
| | 13.00 | 42,860.00 | 17.00 | 21.00 | 4404.69 | 8368.91 | 209.75 | 4.78 |
| | 14.00 | 42,864.00 | 18.00 | 21.00 | 2251.00 | 4276.90 | 107.19 | 2.44 |
| | 15.00 | 42,878.00 | 12.00 | 21.00 | 1756.68 | 3337.69 | 83.65 | 1.91 |
| | 16.00 | 42,880.00 | 16.00 | 27.00 | 3507.41 | 6664.08 | 129.90 | 3.81 |
| | 17.00 | 42,881.00 | 13.00 | 21.00 | 1708.60 | 3246.35 | 81.36 | 1.86 |
| | 18.00 | 42,885.00 | 14.00 | 27.00 | 3400.79 | 6461.50 | 125.96 | 3.69 |
| | 19.00 | 42,894.00 | 15.00 | 21.00 | 3883.94 | 7379.49 | 184.95 | 4.22 |
| **Line 3 (10 uds.)** | 20.00 | 42,865.00 | 29.00 | 21.00 | 1457.00 | 2768.30 | 69.38 | 1.58 |
| | 21.00 | 42,866.00 | 26.00 | 21.00 | 2507.00 | 4763.30 | 119.38 | 2.72 |
| | 22.00 | 42,867.00 | 28.00 | 21.00 | 1740.00 | 3306.00 | 82.86 | 1.89 |
| | 23.00 | 42,870.00 | 25.00 | 21.00 | 4271.48 | 8115.81 | 203.40 | 4.64 |
| | 24.00 | 42,871.00 | 27.00 | 21.00 | 1830.00 | 3477.00 | 87.14 | 1.99 |
| | 25.00 | 42,874.00 | 20.00 | 21.00 | 1643.00 | 3121.70 | 78.24 | 1.78 |
| | 26.00 | 42,886.00 | 22.00 | 27.00 | 2873.73 | 5460.09 | 106.43 | 3.12 |
| | 27.00 | 42,888.00 | 21.00 | 21.00 | 1547.38 | 2940.02 | 73.68 | 1.68 |
| | 28.00 | 42,891.00 | 24.00 | 27.00 | 2260.50 | 4294.95 | 83.72 | 2.45 |
| | 29.00 | 42,893.00 | 23.00 | 21.00 | 1454.93 | 2764.37 | 69.28 | 1.58 |
| **Line 4** | 30.00 | 42,857.00 | 38.00 | 21.00 | 1293.22 | 2457.12 | 61.58 | 1.40 |
| | 31.00 | 42,857.00 | 36.00 | 21.00 | 2155.26 | 4094.99 | 102.63 | 2.34 |
| | 32.00 | 42,863.00 | 37.00 | 21.00 | 1551.32 | 2947.51 | 73.87 | 1.68 |
| | 33.00 | 42,884.00 | 35.00 | 21.00 | 1919.48 | 3647.01 | 91.40 | 2.08 |
| | 34.00 | 42,892.00 | 32.00 | 27.00 | 2873.76 | 5460.14 | 106.44 | 3.12 |
| | 35.00 | 42,894.00 | 31.00 | 21.00 | 2291.25 | 4353.38 | 109.11 | 2.49 |
| | 36.00 | 42,895.00 | 33.00 | 21.00 | 2832.66 | 5382.05 | 134.89 | 3.08 |
| | 37.00 | 42,898.00 | 34.00 | 21.00 | 1519.43 | 2886.92 | 72.35 | 1.65 |
| | 38.00 | 42,899.00 | 30.00 | 21.00 | 2157.00 | 4098.30 | 102.71 | 2.34 |
| | Total | | | 840.00 | 92,962.93 | 176,629.57 | 110.67 | 100.93 |

Construction of a Reinforced Backfill

In the area where the settlements reflected a greater subsidence, a deeper treatment was carried out (Figure 12), consisting of a 3 m soil remediation, injections, a bidirectional reinforcement (two uniaxial layers), and subsequent reconstruction by means of reinforced fill and a lighter treatment on both sides of this area of greater deformation (following the route of the road).

From a geometrical point of view, for this treatment, the voids between injections were considered voids of a certain diameter, using one or more layers of geogrid to bridge them. These geogrids can bridge the differential settlements that occur, anchoring themselves in fixed areas. The solution adopted was based on this concept. In the case of a generalised settlement in the backfill, where no fixed points existed, the geogrid simply descended in solidarity with the backfill. Therefore, the design was made on the assumption that the area of differential settlement had a defined direction.

The tensile and deformational characteristics of the geogrid was set based on the thickness of the drainage (3 m) and the diameter of the possible cavity (space between injections), which was considered 2 m.

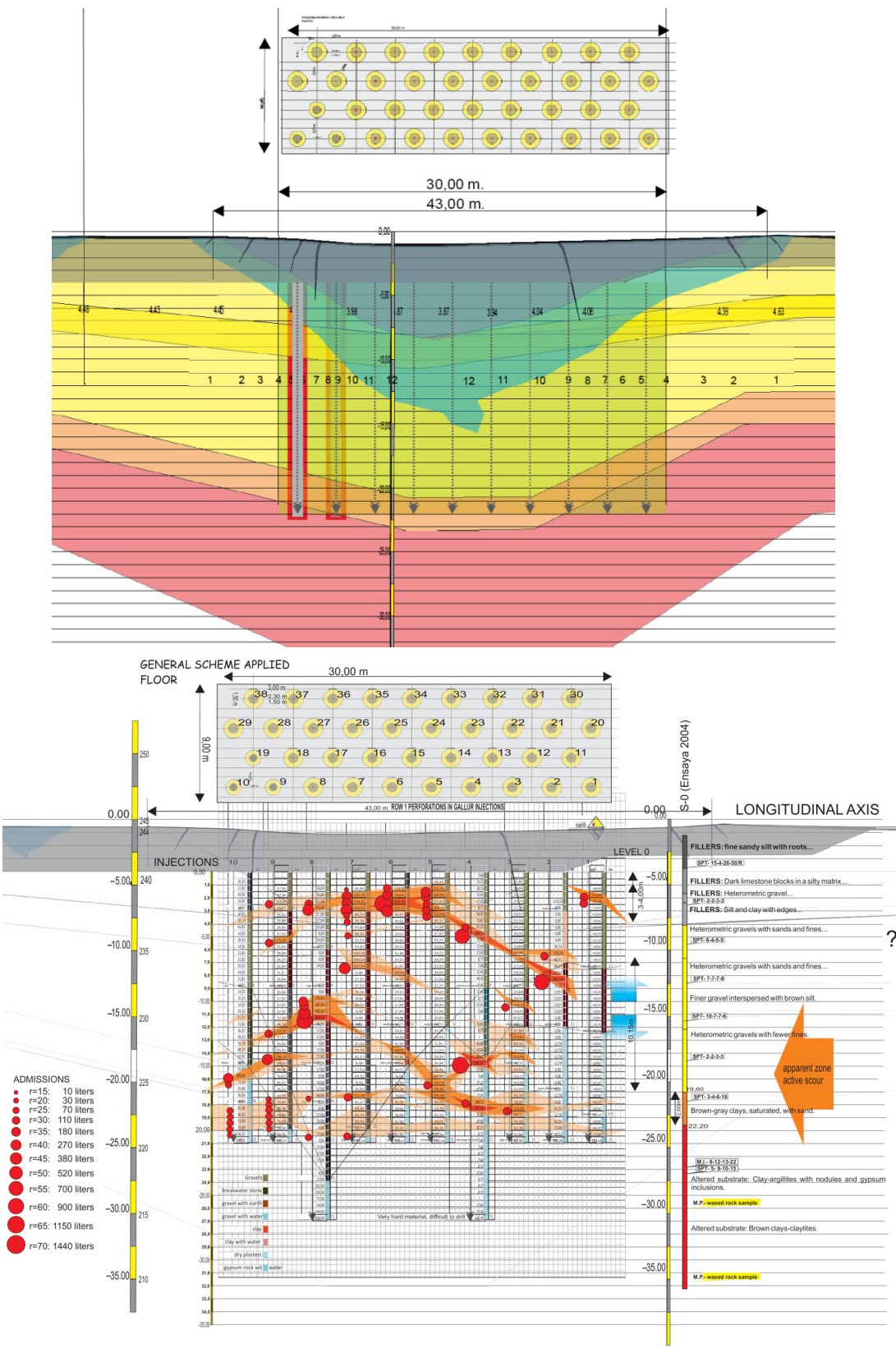

**Figure 11.** Profile showing the arrangement of the injections in the section with the highest apparent subsidence at the surface, where the correlation model showed the largest deformations.

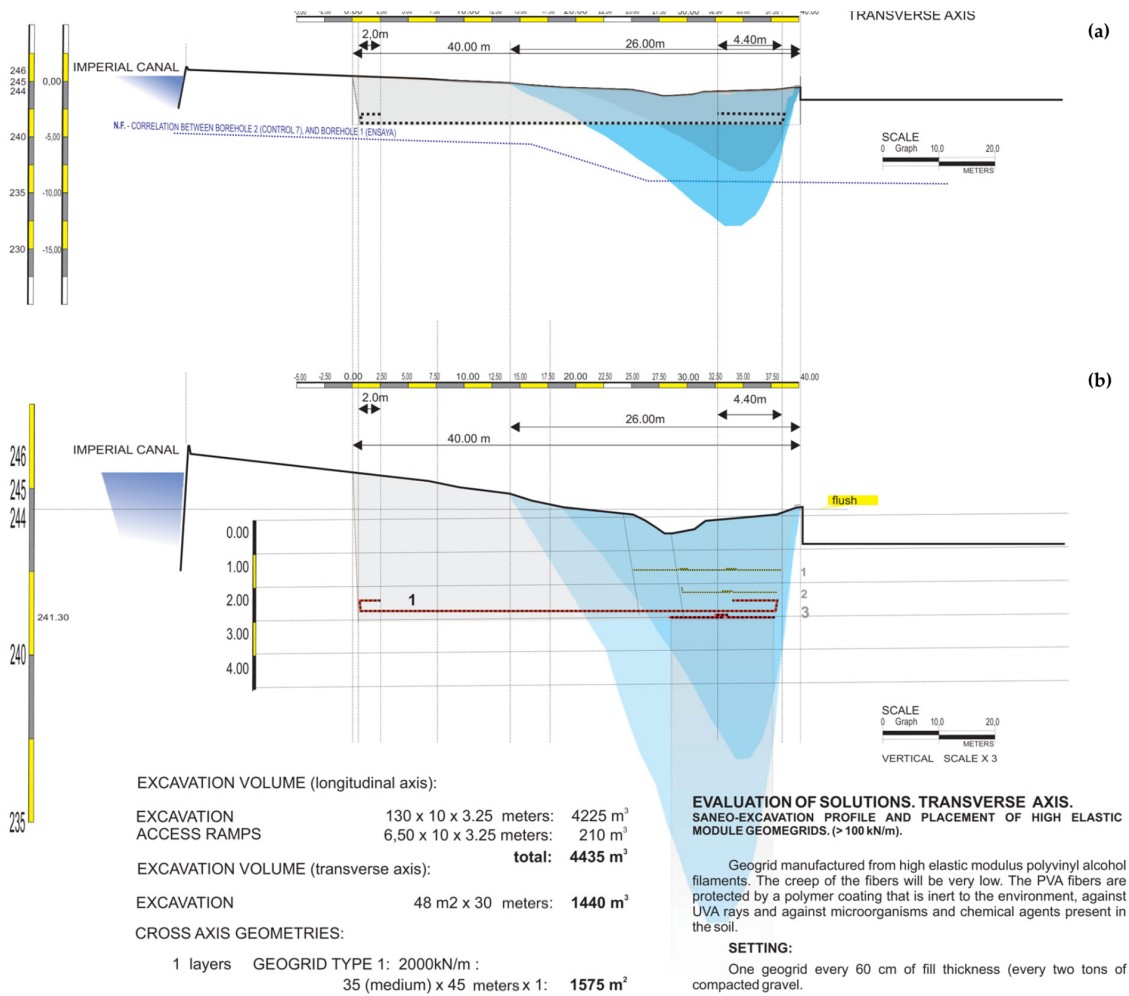

**Figure 12.** Profile of the reinforced backfill solution: (**a**) Longitudinal profile; (**b**) Transverse profile. A series of photographs showing this treatment are attached in Figure 13.

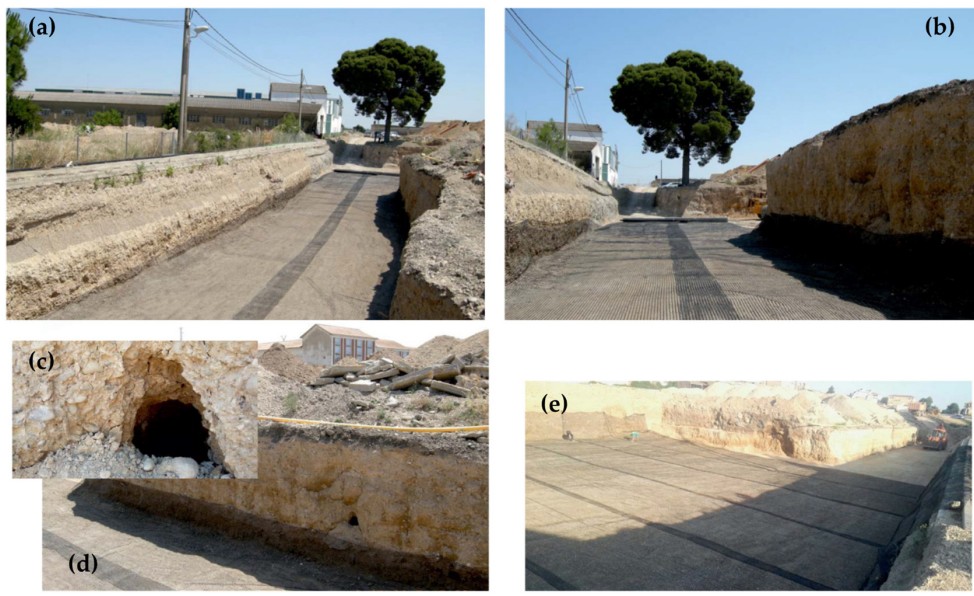

**Figure 13.** Photographs showing part of the reinforced backfill solution shown in Figure 12: (**a**,**b**) Longitudinal profile area; (**c**–**e**) Transverse profile area. (**c**) shows in detail a karstic cavity detected during the treatment.

An H/D ratio < 1 (H being the infill thickness and D the diameter), the British standard [24], was used, although considering the method established by Giroud [25], usually the ratio used is H/D > 1. This distinction is due to the fact that the British standard [24] does not consider the arch effect produced in the infill for H/D ratios > 1, so if the design is carried out with this standard, the results obtained are exaggeratedly on the side of safety. On the other hand, for H/D < 1, as the effect of the discharge arc does not occur, this standard gives solutions that are in line with reality. The method of Giroud [25] takes into account the effect of the unloading arch, which substantially reduces the vertical pressure that the geogrid must bridge, which for H/D ratios > 1 is quite close to reality.

In terms of acceptable surface deformation, following the indications of the British standard [24] for the case of secondary roads, this was set at 2%. It is of great importance in this type of application to use materials whose raw material has a low nominal deformation; for this reason, Polyvinyl Alcohol (PVA) was chosen, which has a nominal deformation <6%.

The characteristics of the geogrids used were as follows:

- Positioned at the base of the excavation (at −3.00 m; layer 5): GEOMALLA woven geotextile based on polypropylene, type FORTRAC (HUESKER brand), made of PVA and with a combined modulus of 2000 kN/m.
- Positioned in 60 cm intervals (layers 1 to 4): GEOMALLA woven geotextile based on polypropylene, type FORTRAC (HUESKER brand), made of PVA and with a combined modulus of 1000 kN/m.

As for the installation of the geogrid, once the clean-up was performed, the geogrid was laid out on a surface, as smooth as possible and without elements that could damage it, in parallel rolls with a minimum overlap of 0.5 m between them.

As the bridging of the voids was unidirectional, if the length of the action was greater than the length of the rolls, an overlap corresponding to the anchor length plus the void to be bridged plus the anchor length was carried out. Therefore, the action zone was extended by the anchorage length on each side to ensure the anchorage of the geogrid determined in each zone.

A layer of about 15–20 cm was placed between the two lines of uniaxial layers placed for subsidence bridging at the bottom of the excavation (one in the longitudinal direction and one in the transverse direction).

Finally, the construction of the backfill was carried out with selected granular material, spread, moistened, and compacted in 20–30 cm thick layers (compaction degree 95% of the modified proctor), up to −3.00 m (Figure 14).

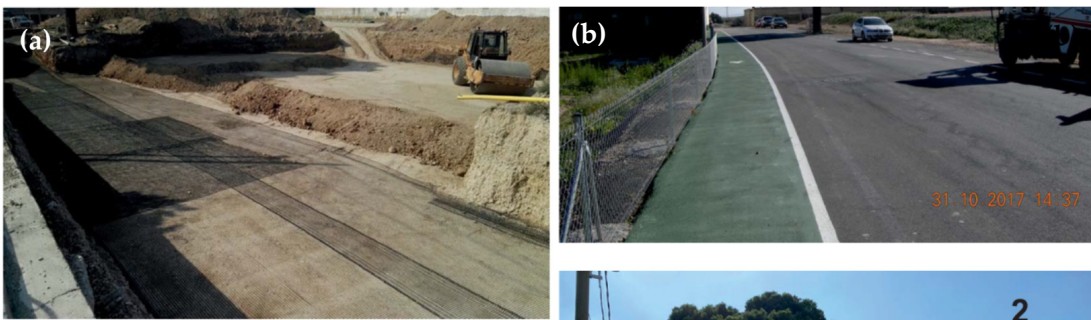

**Figure 14.** (**a**) Construction of the backfill with granular material; (**b**) Final state of the area after treatment and asphalting.

## 4. Conclusions

The modeling of this type of situation, in which the ground sinks and the possible apparent causes may be related to processes that develop at very variable depths, requires an investigation that includes different points of view.

The study of the background is particularly interesting. The stories of many neighbors about events that occurred in previous times often resolves the origin or possible origin of the process in question. In the case of Gallur, there were already known problems of subsidence that affected a house located at the specific point where this work is focused and a section of the Canal Imperial de Aragón, which is in the vicinity. Previous geotechnical studies already existed in which the problem was analyzed and included soundings and other types of tests and investigations, mainly geophysics.

Based on the geomorphological analysis of the ground surface, it was already possible to recognize the subsidence that affected mainly one of the main access roads (Camino Real) to the town of Gallur. The detailed topography of the area and its immediate surroundings allowed us to point out the extent and location of the depocenters of a main subsidence and at least two of lesser magnitude.

Using the rotary borings with continuous core drilling, carried out in previous campaigns, a geological and geotechnical model was elaborated, adjusted fundamentally to the access road indicated. It was possible to evaluate the layers of the recharge ground, the situation of the rocky substratum, and the local water table. It was possible to appreciate the disposition of the different levels or strata in the form of a basin, pointing out the apparent subsidence process affecting the terrain and reflected on the surface.

Based on all the existing data, it has been possible to propose sustainable, novel, and efficient solutions that allow us to continue using the infrastructure in a safe way and that can be useful to alleviate similar problems in other types of situations similar to the one studied in this work.

The adopted solution entailed two key measures: consolidating the ground in the settlement-affected section through the injection of low-mobility mortar at the sunken area's core and constructing a reinforced embankment using high-tensile-strength geogrids along the entire affected street section.

When considering alternative solutions, environmental impacts were a significant factor, especially regarding the injection procedure. Hence, the chosen method relies on special mortars categorized as 'low mobility.' This designation implies that they remain localized at the injection site without being carried away by sub-valve flows. These mortars, referred to as "high viscosity dry mortars", do not blend with the soil, nor do they alter its permeability or cause fractures. Instead, they form concentrated masses, injected at extremely high pressures up to 60 bar, filling potential voids and compacting and solidifying loose soil. Consequently, they have minimal interaction with the surrounding environment and remain inert once set, making them a sustainable option.

In comparison, techniques like more liquid injections or jet grouting systems utilize fluid products that can integrate more readily into underground flows or seepage, rendering them significantly more reactive than the low-mobility mortar method.

**Author Contributions:** Conceptualization, A.G., F.J.T. and J.G.-R.; methodology, A.G. and F.J.T.; software, A.G. and M.P.-P.; validation, A.G., F.J.T. and J.G.-R.; formal analysis, A.G., M.P.-P. and J.G.-R.; investigation, A.G., M.P.-P. and F.J.T.; resources, A.G. and M.P.-P.; data curation, A.G. and F.J.T.; writing—original draft preparation, A.G.; writing—review and editing, F.J.T., J.G.-R. and M.P.-P.; visualization, F.J.T., J.G.-R. and M.P.-P.; supervision, F.J.T. and J.G.-R.; project administration, F.J.T. All authors have read and agreed to the published version of the manuscript.

**Funding:** This research received no external funding.

**Institutional Review Board Statement:** Not applicable.

**Informed Consent Statement:** Not applicable.

**Data Availability Statement:** The data may be available on request from the first author but are not publicly available due to being private.

**Acknowledgments:** We would like to thank the City Hall of Alcalá village for providing permissions and logistic support during the measurement campaigns. Thanks are also given to the Environmental



Management Area of the Ebro Hydrographic Confederation (CHE). We are also grateful to Jesús Rico (Associated Technical Consultants, CTA, S.A.P.) for his collaboration in the geotechnical reports.

**Conflicts of Interest:** The authors declare no conflicts of interest.

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
