# Peer review of "Identification and Mitigation of Subsidence in Karstic Areas with Sustainable Geotechnical Structures: A Case Study in Gallur (Spain)"

_sustainability, doi:10.3390/su16093643_

Round 1
Reviewer 1 Report
Comments and Suggestions for Authors
Overall, a very interesting, well presented and well reasoned paper. But please check the technical English, some geotechnical jargon words are badly translated by whatever system you have used for this purpose. I give you some examples in the Quality of English Language section. Congrats!
Comments on the Quality of English LanguageIn general, it can be said that the level of English in this paper is more than adequate, clearly conveying the ideas it wishes to express. It is likely that by using some English correction or enhancement tool, mistranslations have slipped in, as in line #17 where settlement has been translated as seating, or leakege/seepage as filtration . A careful reading of the manuscript by the authors should eliminate such cases.
Reviewer 2 Report
Comments and Suggestions for Authors
The manuscript describes an interesting case study of sinkhole formation due to dissolution processes in gypsum soils near Zaragoza (Spain). The work is informative and the field study provides useful data of subsidence and observed consequences in surrounding infrastructure (buildings and roads). The paper is well conceived although it requires some editorial work to improve its readability (specially in the abstract) as there are some grammatical errors and some level of repetition which could be avoided.
Specific comments are listed below:
Abstract: please avoid repetition, specially towards the middle part of the paragraph. Sentence “For this reason, the road has been closed for several years” is odd; which road? It is suggested that the last sentences clarify clearly the main content of the paper (case study). Please revise the text to tiddy up the main ideas.
Introduction: line 82, in what sense were the remediations “sustainable”. Did they address any of the three pillars of sustainability: economy, environment and society. If so, how?
Section 2.1 is well written. This section is highly descriptive. Was the information from National Geological Survey (e.g. IGME) complemented by other sources? The text refers to some “spectacular collapses”, are there any of these evidences on record or described in the literature? Is this the only paper describing cases studies in this region?
Fig. 6: please provide which “existing data” was used.
Line 217: the solution of injecting low-mobility mortar is interesting. How did the authors consider potential environmental impacts in balancing alternative solutions.
Conclusions: overall this section is reasonable although some additional support is needed in the manuscript to claim that the solution is “sustainable”. Perhaps this is something that could be added in the review. The novelty aspects are highlighted.
Comments on the Quality of English Language
The manuscript needs a bit of editing to improve its readability (specially in the abstract).
Reviewer 3 Report
Comments and Suggestions for Authors
Dear Authors,
I have no doubt that this is an interesting case study. However, the paper has shortcomings that need to be addressed to make this research publishable. For example, there is a lack of information on the characterization of the terrain. Neither the techniques and procedures, nor their location in the study area are described. It would be very convenient to have more details on the exploration of the terrain. Likewise, it would be advisable to provide geotechnical data. On the other hand, in the discussion of results there is no reference bibliography or comparable cases, which does not support the conclusions.
In any case, there is one aspect that can necessarily be improved, and that is the graphic section. Figure 1 has colors, but there is no data on their numerical meaning, there is no scale or quantification. Figure 2 contains unreadable information. My advice is to merge them and simplify the content, for example, by redrawing part of the curves. In the current state neither the isolines nor the colors are identified. Figure 3 lacks scale data (e.g., graphic scale), and it would be desirable to have coordinates in the aerial image as well.
In general, there is a problem with figures that do not correspond to photographs.Most of them contain information, such as labels, whose content cannot be seen. This is probably the result of using figures from geotechnical reports. It is essential to redo them so that their content adequately illustrates the text, which is not currently the case. This is essential for the reader to be able to judge the quality of the research presented.
I encourage the authors to take into account these recommendations, which are made from a constructive point of view, and are approachable.
Round 2
Reviewer 3 Report
Comments and Suggestions for Authors
The authors have made modifications that improve the manuscript, in accordance with the suggestions made during the review. This research is ready for publication.